# The Impact of Past COVID-19 Infection on Selected Lymphocyte Subsets in Pediatric Patients

**DOI:** 10.3390/vaccines11030659

**Published:** 2023-03-15

**Authors:** Oskar Budziło, Ninela Irga-Jaworska, Marcelina Myszyńska, Magdalena Malanowska, Maciej Niedźwiecki

**Affiliations:** Department of Paediatrics, Haematology and Oncology, Medical University of Gdańsk, M. Skłodowskiej-Curie 3a Street, 80-210 Gdańsk, Poland

**Keywords:** COVID-19, lymphocyte, pediatric, flow cytometry

## Abstract

The impact of past COVID-19 infection on the immune system remains unidentified. So far, several papers have revealed the dependence between the count of lymphocytes and their subsets and the outcome of an acute disease. However, still there is little information about long-term consequences, particularly in the pediatric population. We attempted to verify whether a dysregulation of the immune system may be the reason for observed complications after past COVID-19 infection. Hence, we tried to prove that abnormalities in lymphocyte subpopulations are found in patients a certain time after the COVID-19 infection. In our paper, we enrolled 466 patients after SARS-CoV-2 infection, and evaluated their subsets of lymphocytes within 2–12 months after infection and compared them to the control group assessed several years before the pandemic. It occurred that main differences are observed in CD19^+^ lymphocytes and the index CD4^+^/CD8^+^ lymphocytes. We believe that this is only the introduction to further investigation of the immune system of pediatric patients post-COVID-19 infection.

## 1. Introduction

The first cases of coronavirus disease 2019 (COVID-19) were reported in December 2019 in Wuhan, China. Since then, the disease has spread all over the world and fulfilled the definition of a pandemic [1]. The new pathogen, belonging to the family of coronaviruses, was named Severe Acute Respiratory Syndrome Corona Virus 2 (SARS-CoV-2). Unfortunately, it obviously occurred that immunity created after contact with other coronaviruses does not refer to the new kind of virus [2,3]. The clinical outcome of the disease varies much between individuals. Patients infected with SARS-CoV-2 may present no symptoms at all, mild common cold-like symptoms, a moderate clinical picture with a need for hospitalization or, finally, a severe form with massive pneumonia in some cases leading to a fatal outcome [4]. Yet it has been observed that especially elder patients with comorbid chronic disorders have a higher risk of death due to COVID-19 [5,6,7,8,9]. Hence, in pediatric patients, the percentage of deaths in COVID-19 is significantly smaller than in adults. Up to 9 August 2022, 583 million cases of COVID-19 were confirmed [WHO] together with 6.4 million deaths. Only 0.4% of death cases are represented by children and adolescents [Unicef]. Still, it would be a mistake to overlook the fact that pediatric patients suffer from short- and long-term complications after the infection, which include loss of taste and smell, fatigue, headache, chest pain, higher susceptibility to respiratory tract infections and specific multisystem inflammatory syndrome in children temporally associated with SARS-CoV-2 (MIS-C) [10,11,12,13,14]. For a better understanding of the post-COVID-19 complications and taking into consideration that in patients during acute phase of the disease a significant leukopenia is observed [15], we decided to investigate the immune status of pediatric patients who were diagnosed with COVID-19, especially regarding lymphocyte subset counts.

## 2. Materials and Methods

As a multi-specialist pediatric center in Northern Poland, since February 2021 we have admitted about half a thousand children after recovery from COVID-19 who were referred to the high-reference hospital by general practitioners due to the suspicion of potential complications. In this retrospective single-center study, we enrolled 466 patients aged 0–18 years. All of the patients were confirmed to undergo COVID-19 by RT-PCR test during the infection or by the documented presence of IgG antibodies against SARS-CoV-2 as long as they were not vaccinated against COVID-19. Children were admitted 2 to 12 months since the coronaviral infection. The admission was performed as the one-day stay. Patients did not present any visible signs of infectious disease at that point. During standard immune diagnostic processes, we analyzed subsets of lymphocytes in peripheral blood: CD3^+^, CD3^+^CD4^+^, CD3^+^CD8^+^, CD19^+^, CD56^+^, and CD3^+^ HLA-DR^+^. The results were presented as a count of each subpopulation using 10^9^/dm^3^ unit (G/L). Further, the percentage out of the total count of lymphocytes was calculated. Index CD4^+^/CD8^+^ was calculated by dividing the count of CD4^+^ lymphocytes by the count of CD8^+^ lymphocytes.

The whole blood samples were processed up to 3 h from venipuncture. Briefly, 100 μL of EDTA whole blood samples were stained with CYTO-STAT tetraCHROME CD45-FITC (clone B3821F4A)/CD4-RD1 (clone SFCI12T4D11)/CD8-ECD (clone SFCI21Thy2D3)/CD3-PC5 (clone UCHT1), or CYTO-STAT tetraCHROME CD45-FITC (clone B3821F4A)/CD56-RD1 (clone N901/NKH-1)/CD19-ECD (clone J3-119)/CD3-PC5 (clone UCHT1), or IOTest CD3-FITC (clone UCHT1)/HLA-DR-PE (clone Immu357) Monoclonal Antibody Mixture (Beckman Coulter; Brea, CA, USA) strictly according to the manufacturer’s recommendation. Samples were then lysed with no wash technique with the IMMUNOPREP Reagent System and TQ-Prep Workstation (Beckman Coulter; Brea, CA, USA). Finally, cells were analyzed with Navios EX Flow Cytometer (Beckman Coulter; Brea, CA, USA) calibrated with the manufacturer’s fluorescence beads (Flow-Check and Flow-Set Fluorospheres). The absolute cell counting was calculated upon lymphocytosis from a CBC assay (Sysmex; Kobe, Japan). The lymphocytes were gated as CD45 high/SSC low, and next, the subpopulation of interest was gated as CD4^+^/CD3^+^ (CD4^+^ T cells); CD8^+^/CD3^+^ (CD8^+^ T cells); CD19^−^/CD3^+^ (total T cells); CD19^+^/CD3^−^ (B cells); and CD56^+^/CD3^−^ (NK cells). The activated T cells were gated as SSC low/FSC low lymphocytes and the next HLA-DR^+^/CD3^+^ double-positive T cells. The cut-offs for each population were established as an FMO (fluorescence minus one) approach. Each patient’s run was under control with the CD-Chex Plus whole blood reference material (Streck; La Vista, NE, USA) and INSTAND, EQAS program (Dusseldorf, Germany).

The results were compared with a control group of 22 healthy children aged 4–16 years. Clinical data and laboratory findings were collected in the medical database constructed in Microsoft Excel software for Windows 10 (Microsoft, Redmond, WA, USA). Data were analyzed using Statistica software version 12.6 for Windows (StatSoft Inc. 2015, Tulsa, OK, USA). Shapiro-Wilk test was used to estimate either normal or abnormal spread of analyzed variables. Depending on the spread of variable, nonparametric Mann–Whitney U test, ANOVA Kruskal-Wallis test, Wilcoxon test, ANOVA Friedman test, and parametric Student’s t-test were used. Chi-square test and estimation of the correlation (R Spearman, Pearson) were used for statistical analysis of some variables. Significance level was *p* < 0.05. Data were presented as mean value, standard deviation (SD) and interquartile range (IQR) [16].

## 3. Results

By comparing the results of the control group with patients who underwent COVID-19, we found only few significant differences. Both the count and the percentage of CD19^+^ lymphocytes were higher in the group who recovered from SARS-CoV-2 infection: 0.70 G/L (IQR 0.32–0.91) vs. 0.45 G/L (IQR 0.24–0.52) and 19.17% (IQR 15–23) vs. 13.36% (IQR 9–17), respectively (Figure 1 and Figure 2). In both cases, *p* value occurred <0.05. Surprisingly, we discovered that the proportion of CD3^+^ lymphocytes (Figure 3) was significantly lower (*p* < 0.05) in the patients group: 68.22% (IQR 64–73) vs. 71.18% (IQR 65–76), while the absolute count of CD3^+^ lymphocytes was higher. This may be explained due to the fact that the total count of lymphocytes was also higher in the patients group. Still, it is necessary to emphasize that the last two differences were statistically irrelevant (*p* > 0.05). The last significant difference was found in the CD4^+^/CD8^+^ index parameter (Figure 4). Here, the index value was significantly higher in patients than in controls: 2.01 (IQR 1.5–2.4) vs. 1.61 (IQR 1.3–1.9). All other differences between the two groups were statistically insignificant.

## 4. Discussion

For a better understanding of our findings, we carried out a brief review of the literature concerning characterization of T lymphocytes in COVID-19 patients. With regard to adults, we found three research articles, one short communication and one meta-analysis. From all papers, it is clear that COVID-19 patients present significant lymphocytopenia during the acute phase of infection. Interestingly, in SARS-CoV-1 and MERS patients, similar findings are observed [17]. Possible theories to explain such phenomena is mobilization of immune cells to sites of infection or direct destruction of T cells by a virus [18].

Fenoglio et al. discovered that the frequency of CD3^+^ lymphocytes was significantly lower in infected individuals, while the proportion of CD4^+^ and CD8^+^ remained comparable with a control group. In sequence, activation markers such as CD38^+^ and HLA-DR^+^ antigens were examined, which revealed a significant increase in memory and effector T-cell subsets. While investigating helper subpopulations, no difference was found in the matter of Th2, Th9, Th17 and Th22 subsets, but the prevalence of both Th1 and Th17-1 subpopulations was lower in patients than controls. Finally, regulatory T cells (Treg) were compared, which resulted in the finding that CD8^+^CD28^-^CD127^low^CD39^+^ Treg frequency was relevantly higher in the patients group. In conclusion, it is emphasized that immune response to SARS-CoV-2 is impaired in the Th1 part while immune mechanisms sustaining inflammatory process together with recruitment of CD8^+^ Tregs may impede the effective immune response [19]. From our results, it seems that after recovery in children the total count of CD3^+^ lymphocytes comes back to normal values, while the proportion remains lower. The last may be connected with higher CD19^+^ count, but the exact consequence of this phenomenon is unclear.

Regarding helper subpopulations, a similar conclusion evolved from the research conducted by Lombardi et al. In the last study, the Th1/Th2 ratio was significantly altered in patients, which indicates that CD4^+^ lymphocytes of COVID-19 patients are more likely to differentiate toward Th2 subpopulations. When confronting the investigated variables with 28-day mortality, it occurred that patients with fatal outcome presented more severe lymphocytopenia and higher frequency of activated lymphocytes with CD38^+^ and HLA-DR^+^ expression. Yet it is still unknown whether the mentioned differences are a natural consequence of severe infection or whether, in these particular patients, the equilibrium between innate immune response and control mechanisms was impaired [20].

Bobcakova et al. differentiated their findings by the severity group. As a result, they discovered that the more severe the course of the disease, the lower the count of lymphocytes observed. Additionally, in the most severe groups of patients even complete depletion of CD19^+^ lymphocytes was observed. As the recovery process proceeded, the count of lymphocytes and their subsets showed a tendency to normalize. On the contrary, in *Fenoglio* et al. a significantly lower proportion of CD38^+^HLA-DR^+^CD8^+^ was observed in extremely severe patients, while proportion of CD38^+^CD8^+^ was higher. Interestingly, the presence of exhaustion marker PD-1 both on CD3^+^CD4^+^ and CD3^+^CD8^+^ cells correlated with fatal outcome. The last finding fits the conclusion of Wang et al. [15,19,21]. Yet, while investigating the relevance of Tim-3 expression on CD3^+^CD4^+^ and CD3^+^CD8^+^ cells, no significant differences were found. Hence, the expression of CD38^+^ on CD3^+^CD8^+^ cells with or without simultaneous expression of PD1 on CD3^+^CD4^+^ cells may be utilized as a biomarker of a fatal outcome in hospitalized patients with COVID-19 [15]. A meta-analysis conducted by Huang et al. confirms the reports mentioned above and links the intensity of depletion with severity of the clinical course of COVID-19. Interestingly, the depletion in CD19^+^ and CD56^+^ was less expressed than CD4^+^ and CD8^+^ [22]. In our study, we proved that in recovered pediatric patients neither the depletion of CD19^+^ nor HLA-DR^+^ lymphocytes were observed.

So far, mentioned studies have not focused on atypical lymphocytes (AL). They are described as non-malignant, morphologically changed T cells which by their shape resemble a scallop. They are characteristically bigger and present an active DNA synthesis. Typically, AL are found in patients with EBV infection, but may also occur in cytomegaly, rubella, herpes, varicella and mumps [23]. Sugihara et al. investigated the correlation between a presence of AL and clinical course of COVID-19. It was found that the patients who were confirmed to have circulating AL in the peripheral blood were more likely to present radiographic pathologies suggesting pneumonia and more often needed oxygen-based therapy. On the other hand, two-third of these patients presented a significant improvement shortly after the appearance of AL. Interestingly, it seems that among other respiratory tract infections, COVID-19 is unique in the matter of prevalence of AL. Still, it needs to be emphasized that AL were found only after about a week since the onset of the disease; hence, they cannot be useful for early diagnosis of COVID-19 [24].

While in adults there are multiple research studies concerning immune status of COVID-19 patients, only few reports refer to pediatric patients. Mahmoudi et al. described similar dependence between the count of lymphocytes and the severity of clinical course in children suffering COVID-19. Nevertheless, only 34.5% of analyzed patients presented lymphopenia at all. The group with severe pneumonia was identified with significantly lower count of lymphocytes than the mild/moderate group. Furthermore, in severe cases increased count of CD8^+^ T cells and lower percentage of CD4^+^ T cells were observed, but those differences were statistically negligible. Another relevant difference was identified in CD4^+^/CD8^+^ index, which was distinctly lower in the severe group. Such pathology is mainly observed in HIV and correlates with an advancement of the disease. Still, in COVID-19 patients a significance of this finding is unclear [25]. On the contrary, *Argun* et al. did not find any differences neither in total lymphocyte count nor in CD4^+^/CD8^+^ index between mild/asymptomatic COVID-19 pediatric patients and controls. However, natural killer T and CD4^+^ T cell counts were significantly higher in patients than in controls. That may indicate children have an ability to react with proliferation of certain subsets of lymphocytes and thereby limit the risk of poor outcome [26]. Yet in our paper, the imbalance between CD4^+^ and CD8^+^ persisted, although the vast majority of patients presented with either a mild or moderate course of the disease.

In our study, we focused more on long-term effects on immune system of children who recovered from COVID-19. To our knowledge, this is the first study assessing such a large group of children with a history of SARS-CoV-2 infection. The point of interest was to help to explain observed clinical complications mentioned in the introduction. The lower percentage of CD3^+^ lymphocytes in patients corresponds with higher proportion of CD19^+^ lymphocytes, so taking into consideration that there was no significant difference in total count of CD3^+^ lymphocytes between the groups, it seems reasonable to neglect this outcome. Surprisingly, the CD4^+^/CD8^+^ index is relevantly higher in patients than in controls. Similar findings are observed in patients with sarcoidosis [27] and in lymphoproliferative diseases [28]. High index is also proposed to be a marker of poor outcome of adult COVID-19 patients [29,30]. Corresponding to the complications observed after coronavirus infection, it is worth to mention that CD4^+^ lymphocytes exert interleukin 4, 5 and 12, which have a positive influence on proliferation of lymphocyte B [31,32]. Hence, the advantage of CD19^+^ lymphocytes may be observed. Additionally, the last are connected with allergic disorders and oversensitivity. Further, most T cytotoxic lymphocytes present CD8 antigens [33,34]; in that case, higher incidence of infections could be explained. If this was not enough, regulatory T cells likewise mostly present CD8 antigens. At present, it has been proven that disproportion in the matter of Tregs is connected with autoimmune diseases. It cannot be omitted that among complications after COVID-19 infection, such symptoms as Reynaud symptom, rush and hives are observed. On the other hand, when taking headache and smell disorders into consideration, it is hard to find the connection between immune status of a patient and the prevalence of these symptoms. Supposedly, these abnormalities are the effect of direct action of the virus against the olfactory neurons. It would be a mistake to overlook the fact that the constant feeling of fear and uncertainty as well as isolation may induce anxiety disorders; as the consequence, we may observe chest pain and headaches as the somatization of that. On the contrary it cannot be excluded that induction of auto-antibodies may have influence on nervous system functioning.

In order to gather more specific data and answer the doubts mentioned above, the cytokines profile should be assessed, as well as Tregs and Bregs populations. In our work we assessed only basic subsets of lymphocytes, which obviously is not enough but gives a glance at the general status of the immune system in pediatric patients past COVID-19 infection. Additionally, if the patients were divided by age some other findings could be made, but taking into consideration the small count of the control group, it would be impossible to achieve statistically relevant results. If it was not enough, assessment of comorbid disorders might be taken into account, but in that case we would lose the view of the pediatric population as a whole. To summarize, to date we may state that lymphopenia observed during the acute infection caused by SARS-CoV-2 subsides but the imbalance is observed instead.

## Figures and Tables

**Figure 1 vaccines-11-00659-f001:**
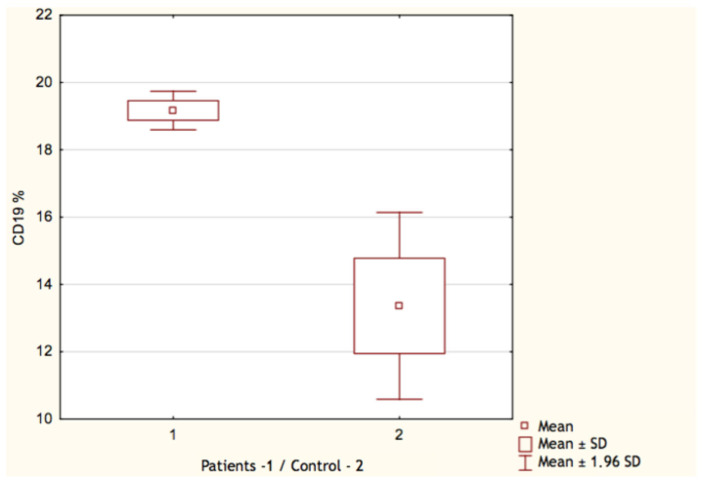
Percentage of CD19^+^ lymphocytes in patients (1) and control group (2).

**Figure 2 vaccines-11-00659-f002:**
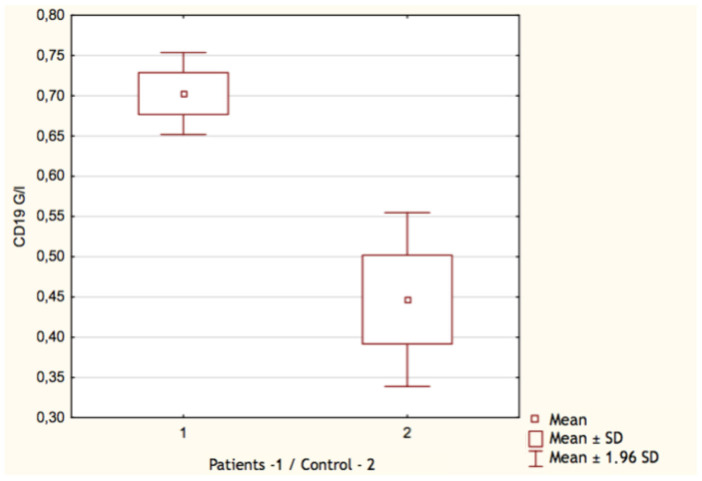
Count of CD19^+^ lymphocytes in patients (1) and control group (2).

**Figure 3 vaccines-11-00659-f003:**
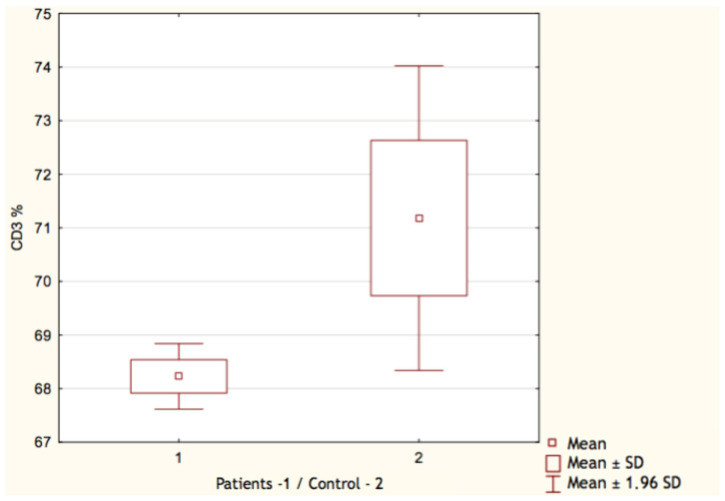
Percentage of CD3^+^ lymphocytes in patients (1) and control group (2).

**Figure 4 vaccines-11-00659-f004:**
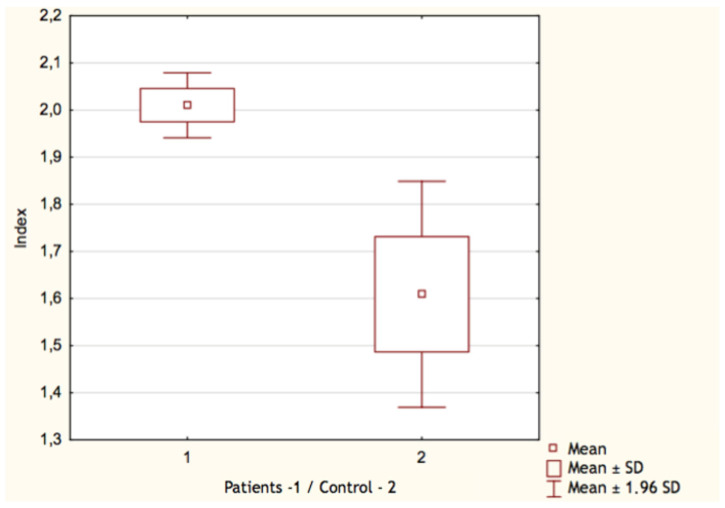
Index CD4^+^/CD8^+^ lymphocytes in patients (1) and control group (2).

## Data Availability

The datasets generated during and/or analyzed during the current study are available from the corresponding author, B.O., on reasonable request.

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
