# Peer review of "The Impact of Past COVID-19 Infection on Selected Lymphocyte Subsets in Pediatric Patients"

_vaccines, 2023, doi:10.3390/vaccines11030659_

Round 1

Reviewer 1 Report

I think this paper reports a retrospective epidemiological research by examining the lymphocyte subtypes of hospitalized children with COVID-19 and comparing them with those of previously examined healthy children. Their ages of hospitalized children ranged from 0 to 18 years old, their hospitalization period was 2 to 12 months, and their degree of illness was probably very distributed. That is why the authors only made a rough comparison. The rough comparison showed that the CD3+ lymphocyte count was significantly lower than that of healthy subjects, and the ratio of CD4+/CD8+ was higher than that of healthy subjects. But, considering that they were inpatients, it seems to be no surprise.

Since this paper is retrospective, I would suggest that the authors re-analyze the data, for example, by separating 466 into ages, hospitalization periods, or medical conditions and compare lymphocyte subtypes with those of the healthy 20 children, respectively.

Although some papers were explained in the discussion, I think it is not necessary to explain each paper in detail, but the comparison and discussion with the own data of this authors should be the main focus. Regarding to AL, it was not mentioned that data of AL was involved in their analysis or not.

Author Response

Point #1. I think this paper reports a retrospective epidemiological research by examining the lymphocyte subtypes of hospitalized children with COVID-19 and comparing them with those of previously examined healthy children. Their ages of hospitalized children ranged from 0 to 18 years old, their hospitalization period was 2 to 12 months, and their degree of illness was probably very distributed. That is why the authors only made a rough comparison. The rough comparison showed that the CD3+ lymphocyte count was significantly lower than that of healthy subjects, and the ratio of CD4+/CD8+ was higher than that of healthy subjects. But, considering that they were inpatients, it seems to be no surprise.

Response #1. The patients enrolled to the study were not inpatients. They were  children recovered from the COVID-19 infection and were admitted to a daily-care ward due to some mild to moderate complaints which were not an indication for a hospitalization. That will be added to the paper. 

Point #2. Since this paper is retrospective, I would suggest that the authors re-analyze the data, for example, by separating 466 into ages, hospitalization periods, or medical conditions and compare lymphocyte subtypes with those of the healthy 20 children, respectively.

Response #2. As the healthy control group counts only 22 children, which is an important limitation we cannot divide the patients group into ages and compare it with adequately divided control group. The vast majority of patients did not need hospitalization during the acute phase of COVID-19 infection. Additional medical conditions could be taken into consideration and in the future and further evaluation of the group we will consider such distinction. I will add this information to the limitations in the last paragraph.

Point #3. Although some papers were explained in the discussion, I think it is not necessary to explain each paper in detail, but the comparison and discussion with the own data of this authors should be the main focus. Regarding to AL, it was not mentioned that data of AL was involved in their analysis or not.

Response #3. We tried to give a short review of the lymphocytes abnormalities among patients during acute phase of infection and collate it with the findings in patients recovered from COVID-19. Indeed some of the mentioned subpopulations were not investigated in our paper, but we cited these findings to give a broader perception of lymphocyte distribution and role in patients suffering from SARS-CoV-2 infection. It will remain as it was initially. 

Reviewer 2 Report

I read it with great interst, but I have raised several concerns.

#1. Since then the disease has spread all over the world and fulfilled the  definition of a pandemi-> Please cie the COVID-19-related article. DOI: https://doi.org/10.54724/lc.2022.e10

#2. Please add the hypothesis of your study.

#3.  In statistical part, please add the statistical guidelines DOI: https://doi.org/10.54724/lc.2022.e1

#4. Please add the limitations in detail.

#5. This is an excellent paper. Thank you.

Author Response

#1 

Since then the disease has spread all over the world and fulfilled the  definition of a pandemi-> Please cie the COVID-19-related article. DOI: https://doi.org/10.54724/lc.2022.e10

# response 1

It will be cited

#2

Please add the hypothesis of your study

# response 2

The hypothesis claims that there are changes in immune status of patients after COVID-19 infection that may bring some explanations to the origin of the complications observed in these patients. It will be more emphasized in the paper. 

#3

In statistical part, please add the statistical guidelines DOI: https://doi.org/10.54724/lc.2022.e1

#response 3

It will be cited

#4

Please add the limitations in detail.

# response 4

Most limitations were given in the last paragraph. Additionally the count of patients in control group makes it impossible to divide the groups by the age of patients, what might show some other conclusions. Moreover we did not divide the patients by the severity of the acute phase of infection due to the fact that only single individuals needed hospitalization during the disease. The vast majority presented mild course of the disease.

#5 This is an excellent paper. Thank you.

# response 5

Thank you.

Reviewer 3 Report

The manuscript is focusing on lymphocyte analysis in patients after SARS-CoV-2. The main focus was on the CD19+, CD3+, and CD4+/CD8+ lymphocyte populations as presented in Figures 1-4. The manuscript is very simple and easy to follow. However, the materials and methods should be updated to include information that would allow the reproducibility of the studies, e.g. description of antibodies and other reagents used, including the catalog numbers. Supplementary information seems to duplicate the Figures and in this case needs to be removed. 

Specific points.

1. Kindly check if the word "select" in the title is appropriate, and whether "selected" could be an option.

2. Line 16: "CD19+". Line 17: "CD4/CD8". Should it be "CD4+/CD8+" for consistency? Also, kindly check the entire manuscript for consistent spelling of lymphocytes with various surface markers. 

3. Line 32: "adults" or "in adults"?

4. Lines 32-33: "up to August 9th". August 9th, kindly add the year. It is nearly February 2023, kindly update and clarify this information.

5. Material and Methods. a) The description of antibodies used for flow cytometry analyses should be more detailed, with indicated fluorophores, producers (company, city, country), and catalog numbers. b) Moreover, it should be described how the lymphocytes were prepared starting from a blood sample point. c) Were there other reagents used, in addition to the antibodies detecting various lymphocytes? E.g., PBS, blocking reagents? Kindly add a description of those as well, including the catalog numbers and producers' information.

6. Line 68. It should be introduced and clarified for non-clinicians, what are "G/l" and "IQR", and how was it measured?

7. Figure 1. a) Dot plots would be more informative regarding the distribution and number of cases than simple squares. The same for Figures 2, 3, and 4. b) Percentage of CD19+ cells out of what population? Kindly elaborate in the Figure legend and text what was the 100% here.

8. Figure 2. What is G/l? was the actual number of lymphocytes counted? "G/l" should be introduced in the materials and methods and mentioned in the Figure legends. 

9. Figure 3. Percentage of CD3+ cells out of what population? Kindly elaborate.

10. Figure 4. What is the "index"? Kindly introduce how was it measured. 

11. Line 94. "SAR-CoV-1". Should it be "SARS-CoV-1"?

12. Line 165. " SARS-Cov-2" should be " SARS-CoV-2"

13. Do supplementary files simply copy the Figures? Then they should be removed, together with lines 197-200.

14. References 15 and 16. Why there are two dots before the first author's name? ". ."

Author Response

  1. Kindly check if the word "select" in the title is appropriate, and whether "selected" could be an option.

Resp. It will be changed into selected.

2. Line 16: "CD19+". Line 17: "CD4/CD8". Should it be "CD4+/CD8+" for consistency? Also, kindly check the entire manuscript for consistent spelling of lymphocytes with various surface markers. 

Resp. It will be changed as suggested.

3. Line 32: "adults" or "in adults"?

Resp. In adults. It will be changed.

4. Lines 32-33: "up to August 9th". August 9th, kindly add the year. It is nearly February 2023, kindly update and clarify this information.

Resp. It refers to 2022 it will be changed.

5. Material and Methods. a) The description of antibodies used for flow cytometry analyses should be more detailed, with indicated fluorophores, producers (company, city, country), and catalog numbers. b) Moreover, it should be described how the lymphocytes were prepared starting from a blood sample point. c) Were there other reagents used, in addition to the antibodies detecting various lymphocytes? E.g., PBS, blocking reagents? Kindly add a description of those as well, including the catalog numbers and producers' information.

Resp. The whole paragraph will be updated with more specific data.

6. Line 68. It should be introduced and clarified for non-clinicians, what are "G/l" and "IQR", and how was it measured?

Resp. G/l means 10^9 pre one liter. As symbol "l" is not universal in some countries I will change it into dm3

7. Figure 1. a) Dot plots would be more informative regarding the distribution and number of cases than simple squares. The same for Figures 2, 3, and 4. b) Percentage of CD19+ cells out of what population? Kindly elaborate in the Figure legend and text what was the 100% here.

Resp. As there is a big patients group we believe that dot plots would not be clear. Percentage of the whole lymphocyte population - it will be added.

8. Figure 2. What is G/l? was the actual number of lymphocytes counted? "G/l" should be introduced in the materials and methods and mentioned in the Figure legends. 

Resp. As mentioned above, it will be added in materials and methods.

9. Figure 3. Percentage of CD3+ cells out of what population? Kindly elaborate.

Resp. As above.

10. Figure 4. What is the "index"? Kindly introduce how was it measured. 

Resp. It means the count of CD4+ lymphocytes divided by the count of CD8+ lymphocytes - it will be mentioned.

11. Line 94. "SAR-CoV-1". Should it be "SARS-CoV-1"?

Resp. It should be SARS-CoV-1.

12. Line 165. " SARS-Cov-2" should be " SARS-CoV-2"

Resp. It should be SARS-CoV-2.

13. Do supplementary files simply copy the Figures? Then they should be removed, together with lines 197-200.

Resp. Indeed. That was misunderstanding. 

14. References 15 and 16. Why there are two dots before the first author's name? ". ."

Resp. Simple mistake - it will be removed. 

Round 2

Reviewer 1 Report

The authors said that only rough comparisons can be made due to the low N of the control group. But, this is not scientific and should not be published as a original paper but as like a letter to authors. Since this study is a retrospective epidemiological study, the results of various classifications and analyses should be presented in the Supplement and integrated to make conclusions.

Again, the authors should explain the papers mentioned in the Discussion and the research of the authors in a one-to-one correspondence. Otherwise the main part of this submitting manuscript would be out of focus.

Author Response

#1 The authors said that only rough comparisons can be made due to the low N of the control group. But, this is not scientific and should not be published as a original paper but as like a letter to authors. Since this study is a retrospective epidemiological study, the results of various classifications and analyses should be presented in the Supplement and integrated to make conclusions.

#Response1

As we plan to to broaden the spectrum of cytokines and subsets of the lymphocytes in the future we will conduct more thorough analysis with the new data then. 

#2 Again, the authors should explain the papers mentioned in the Discussion and the research of the authors in a one-to-one correspondence. Otherwise the main part of this submitting manuscript would be out of focus.

#Response2

We tried to refer more accurately to presented publications.

Reviewer 2 Report

This is an excellent paper.

Author Response

Thank you.